# Effect of Caloric Restriction on BMI, Gut Microbiota, and Blood Amino Acid Levels in Non-Obese Adults

**DOI:** 10.3390/nu12030631

**Published:** 2020-02-27

**Authors:** Hua Zou, Dan Wang, Huahui Ren, Kaiye Cai, Peishan Chen, Chao Fang, Zhun Shi, Pengfan Zhang, Jian Wang, Huanming Yang, Huanzi Zhong

**Affiliations:** 1BGI Education Center, University of Chinese Academy of Sciences, Shenzhen 518083, China; zouhua@genomics.cn (H.Z.); zhangpengfan@genomics.cn (P.Z.); 2BGI-Shenzhen, Shenzhen 518083, China; wangdan6@genomics.cn (D.W.); renhuahui@genomics.cn (H.R.); caikaiye@genomics.cn (K.C.); chenpeishan@genomics.cn (P.C.); fangchao@genomics.cn (C.F.); shizhun@genomics.cn (Z.S.); wangjian@genomics.cn (J.W.); yanghuanming@genomics.cn (H.Y.); 3Shenzhen Key Laboratory of Human commensal microorganisms and Health Research, Shenzhen 518083, China; 4Laboratory of Genomics and Molecular Biomedicine, Department of Biology, University of Copenhagen, 2100 Copenhagen Ø, Denmark; 5James D. Watson Institute of Genome Sciences, Hangzhou 310058, China

**Keywords:** calorie restriction, gut microbiota, enterotype, amino acids, body mass index

## Abstract

Adequate calorie restriction (CR) as a healthy lifestyle is recommended not only for people with metabolic disorders but also for healthy adults. Previous studies have mainly focused on the beneficial metabolic effects of CR on obese subjects, while its effects on non-obese subjects are still scarce. Here, we conducted a three-week non-controlled CR intervention in 41 subjects, with approximately 40% fewer calories than the recommended daily energy intake. We measured BMI, and applied targeted metabolic profiling on fasting blood samples and shotgun metagenomic sequencing on fecal samples, before and after intervention. Subjects were stratified into two enterotypes according to their baseline microbial composition, including 28 enterotype *Bacteroides* (ETB) subjects and 13 enterotype *Prevotella* (ETP) subjects. CR decreased BMI in most subjects, and ETP subjects exhibited a significantly higher BMI loss ratio than the ETB subjects. Additionally, CR induced limited changes in gut microbial composition but substantial microbial-independent changes in blood AAs, including a significant increase in 3-methylhistidine, a biomarker of the skeletal muscle protein turnover. Finally, baseline abundances of seven microbial species, rather than baseline AA levels, could well predict CR-induced BMI loss. This non-controlled intervention study revealed associations between baseline gut microbiota and CR-induced BMI loss and provided evidence to accelerate the application of microbiome stratification in future personalized nutrition intervention.

## 1. Introduction

Calorie restriction (CR), a nutritional intervention of reduced energy intake [1], has shown to be beneficial to reduce body weight, inflammation, and high insulin levels and extend the lifespan of mice [2]. In humans, CR as a healthy lifestyle, is recommended not only for people with metabolic disorders but also for healthy young adults. However, the majority of previous CR-based studies have focused on obese subjects [3,4,5], studies on the effects of CR in non-obese healthy subjects, including potential adverse effects on skeletal muscle, are still scarce [6].

Increasing evidence has demonstrated that body mass index (BMI) is a strong covariate of human gut microbiota. It is clear that obesity is associated with lower gut microbial diversity and richness [7,8]. In turn, the gut microbiota and specific microbial metabolites, such as short-chain fatty acids (SCFA, the end products of fermentation of undigested carbohydrates by the gut microbiota), secondary bile acids, and amino acids and their derivatives, have been implicated in modulating numerous aspects of host energy and metabolism [7,9,10,11,12,13]. For instance, Liu et al. showed that compared to lean controls, young obese adults had a significantly lower abundance of *Bacteroides* spp. and a significantly higher level of plasma glutamate [7]. They further demonstrated that the obese-depleted *Bacteroides thetaiotaomicron*, a glutamate-fermenting microbe, could reduce plasma glutamate levels and alleviate high-fat, diet-induced, body-weight gain in mice [7]. Gut microbiota could also produce imidazole propionate from histidine, which could impair host insulin signaling [13].

Although current studies have consistently shown that CR induces weight loss, findings on CR-induced gut microbial changes in mice and humans are variable. Fabbiano et al. reported that a short-term CR (3–6 week) led to mice gut microbiota alterations, such as a significant reduction in the abundance of Firmicutes and an increase in abundance of Bacteroidetes and Proteobacteria [14]. Zhang and her colleagues showed that a life-long CR intervention could effectively increase gut levels of *Lactobacillus* spp. and extend lifespan in mice [15]. They further revealed a significant enrichment of *Lactobacillus murinus* induced by a 2-week CR in mice [16]. However, Ott et al. reported that a 4-week very-low-calorie diet (800 kcal/day) intervention in obese women did not trigger significant changes in their gut microbial alpha diversity and beta diversity [17]. They suggested that different subjects might have individual-specific microbial responses to the same CR diet, and grouping metagenomic data from all subjects might mask the CR-induced gut microbial changes [17]. Furthermore, the inconsistent CR-related gut microbial changes among studies might also be partially attributed to the differences in experimental design, such as the amount of energy intake and the duration of CR intervention. Therefore, more human research work is needed to elaborate on the underlying mechanisms of CR, gut microbiota, and host metabolism.

On the other hand, an increasing number of studies have revealed that the baseline or pre-treatment stratification of gut microbial composition is associated with different responses to a certain dietary intervention or treatment. Enterotypes, which are driven by dominant gut genera such as *Bacteroides*, *Prevotella*, and *Ruminococcus* [18], have been widely used to stratify populations. Petia et al. reported that a high *Prevotella/Bacteroides* ratio was associated with a beneficial response to barley kernels [19], while Gu et al. revealed that treatment of naïve type 2 diabetes patients with higher abundance of *Bacteroides* showed more considerable improvement in metabolic profiles than those with a higher abundance of *Prevotella*, after three-month acarbose treatment [20]. It is also unclear whether the different gut microbial composition is associated or could even predict host metabolic responses under a given CR intervention.

Here, we applied targeted metabolic profiling and shotgun metagenomic sequencing, separately on blood and fecal samples from 41 non-obese subjects (before and after a three-week CR intervention). Our results revealed that the CR trial could effectively induce weight loss and alterations of host amino acids, but show limited impacts on the gut microbial composition. Despite the absolute daily intakes of protein during CR being substantially lower than regular recommendations, the levels of multiple AAs and their derivatives were significantly increased after intervention in the overall cohort. The significantly elevated level of 3-methylhistidine, a biomarker for skeletal muscle protein turnover, indicated the existence of CR-induced skeletal muscle loss. We further demonstrated that the baseline gut microbial composition (enterotypes or the relative abundances of seven species) could be meaningful predictors of CR-induced BMI loss but was not associated with CR-induced changes of blood AAs. On the contrary, baseline blood amino acid profiling showed no correlation to BMI loss, in response to CR.

## 2. Materials and Methods 

### 2.1. Study Population

Volunteer-wanted posters were propagated at the China National Gene Bank in Shenzhen from March to April 2017. A non-obese healthy volunteer was considered if his/her BMI was less than 28 kg/m^2^. In addition, recruited volunteers had to meet all of the following criteria—(1) no antibiotics intake in the previous 2 months; (2) no prebiotic or probiotic supplement intake in the previous 2 months; (3) not have hypertension, diabetes mellitus, gastrointestinal disease, and other severe auto-immune diseases; (4) regular eating and lifestyle patterns; and (5) no international travel in the previous 3 months. A total of 50 individuals met all criteria and were recruited for the study, and 41 individuals (24 females and 17 males aged 30 ± 6 years old) completed the whole intervention (Table 1). There were no differences in age and BMI between the two sexes and, thus, further analyses were performed on the entire cohort (Table 1). The study was approved by the institutional review board on bioethics and biosafety of BGI-Shenzhen, Shenzhen (NO. BGI-IRB 17020) and registered at clinicaltrials.gov as NCT04044118. All participants were fully informed of the design and purpose of this intervention study and signed a written informed consent letter.

### 2.2. Study Design

The study was designed as an uncontrolled longitudinal study with all volunteers receiving the same intervention but with no control group. Specifically, it included a one-week, run-in period (baseline) and a three-week CR dietary intervention period. During the first week (run-in period), all healthy volunteers consumed their usual diet and were encouraged to avoid yogurt, high-fat foods, and alcohol. During the three-week CR dietary intervention, five different types of low-calorie meals were provided for the five-day workweek (from Monday to Friday), each consisted of 3 meals per day (breakfast, lunch, and dinner). All participants were subjected to the same dietary calorie restriction and were required to finish the meals at the canteen. As no standardized meals were provided at the weekend, all participants were required to take pictures of their food, record, and follow the low daily calorie intake by using the Boohee APP, a mobile application with calorie-counter and food guides. BMI data, fasting blood samples, and fecal samples of each volunteer were collected at our study center at baseline and after the 3-week CR intervention (Figure 1). To avoid intra-individual variations, BMI was measured multiple times for each volunteer, during the last week of the CR intervention, and the averaged BMI value was used as his/her after-intervention BMI (Figure 1). 

### 2.3. Diets

The CR diet consisted of ~60% calories of the recommended daily calorie intake for men and women in the 2016 Dietary Guidelines for Chinese Residents [21] (men: 2400 kcal/day; women: 2000 kcal/day). The average daily calorie supply in this study was 1414.9 kcal/day for men and 1210.6 kcal/day for women, with 43% calories from carbohydrates, 25% calories from protein, and 32% calories from fat (Appendix A). The energy intake per day of the CR diet was calculated based on the detailed composition of the low-calorie meals, using the Chinese food composition tables [22]. Common foods in low-calorie diets such as rice, vegetables, eggs, pork, and beef were prepared in our study center to control the experimental variables introduced by different foods and calorie estimation errors. Traditional Chinese cooking methods—boiling, stir-frying, and stewing—were applied for our diets. For each meal, digital scales were used to measure the nutritional and caloric values of different foods and the total meal for men and women, respectively.

### 2.4. Fecal Sampling and Shotgun Metagenomic Sequencing

Fecal samples were self-collected and then transferred to the laboratory on dry ice and kept frozen at −80 °C before and after the CR intervention. Fecal DNA was extracted, following a manual protocol, as described previously [23]. The DNA concentration was estimated by Qubit (Invitrogen). Library construction and shotgun metagenomic sequencing were performed on qualified DNA samples, based on the BGISEQ-500 protocol in the single-end 100 bp mode [24].

### 2.5. Metagenomic Analysis

Raw reads of BGISEQ-500 with the SE100 mode were trimmed by an overall accuracy (OA) control strategy, to control quality [24]. After trimming, on average, 98.15% of the raw reads remained as high-quality reads (Appendix A). By using the SOAP2.22 software, the high-quality reads were aligned to hg19 to remove the reads from the host DNA (identity ≥ 0.9). The retained clean reads were aligned to the integrated non-redundant gene catalog (IGC) using SOAP2.22 [25], and the average mapping rate and unique mapping rate were 80.18% and 65.76%, respectively (identity ≥ 0.95, Appendix A). The relative abundance profiles of genes, genera, species, and Kyoto Encyclopedia of Genes and Genomes orthologous groups (KEGG, KOs) of each sample were calculated by summing the relative abundances of their assigned IGC genes [25].

For enterotyping, we applied a recently published universal classifier (http://enterotypes.org/), which circumvents major shortcomings in enterotyping methodology, such as lack of standard and small sample size [26]. At the baseline, 41 individuals were clustered into two groups—28 ETB (*Bacteroides* enriched) and 13 ETP (*Prevotella* enriched) individuals. A total of 87.8% (36 of 41) individuals were clustered to the same enterotype, after the three-week CR intervention. Detailed enterotype information for each individual is provided in Appendix A.

Genus or species with an occurrence rate > 80% and a median relative abundance > 1 × 10e-6 in all samples, were defined as common genus or species and used for further intra- and inter-enterotype comparison analyses.

Differentially enriched KEGG pathways were identified between the enterotypes based on the distribution of Z-scores of all KOs belonging to a given pathway [27,28]. A reporter score |Z| > 1.96 (95% confidence interval according to a normal distribution) was used as a detection threshold for significantly differentiating pathways. Alpha diversity of each individual was calculated on the gene and species relative abundance profiles, using the Shannon index. Beta-diversity on the gene and species relative abundance profiles was calculated using the Bray–Curtis distance.

### 2.6. Blood Sample Collection and Amino Acids Profiling

Fasting blood samples were collected before and after the intervention and stored at −80 °C for assessing the effect of CR on host amino acid profiles. The concentrations of 31 amino acids and derivatives in the serum samples were measured via ultra-high performance liquid chromatography (UHPLC), coupled to an AB Sciex Qtrap 5500 mass spectrometry (AB Sciex, Massachusetts, USA), as described previously [29].

### 2.7. Statistical Methods

Pearson’s chi-square test was performed to assess sex distribution between individuals of two enterotypes. Wilcoxon rank-sum test was used to detect the significant differences in phenotypes, the concentrations of blood amino acids, and the relative abundances of genera and species between enterotypes. Wilcoxon signed-rank test was used to detect the significant differences in phenotypes, the concentrations of blood amino acids, and the relative abundances of genera and species in paired samples, before and after the intervention. BMI loss ratio of a given individual was calculated using the following equation:$$BMI\;loss\;ratio=\frac{\;Before_{BMI}-After_{BMI}}{\;Before_{BMI}}*100\%$$
where $Before_{BMI}$ and $After_{BMI}$ are the BMI value of the same individual before and after the CR intervention, respectively. 

The associations between the changes of blood amino acids and the overall baseline gut microbial composition were assessed using permutational multivariate analysis of variance (PERMANOVA) with 9999 permutations on enterotypes (R *vegan* package, *adonis* function). 

Principal coordinate analysis (PCoA) of fecal samples was performed, based on the relative abundances of common species using the Bray–Curtis distance (R *ape* package). Principal component analysis (PCA) was performed based on the blood amino acid profiles to visual overall amino acid composition between enterotypes and between different time points.

To investigate whether we could predict BMI loss ratio using omics features, we performed a Lasso (Least absolute shrinkage and selection operator) regression analysis between baseline relative abundances of common gut species and the concentrations of blood amino acids (independent variables), and BMI loss ratio (dependent variables).

We first normalized the values of both independent and dependent variables (R, *scale* function). We then used the R function *cv.glmnet* to choose the most appropriate value for λ in the Lasso model (R *glmnet* package, family = “gaussian”, nfolds = 10, alpha = 1, nlambda = 100). Here, λ is the tuning parameter (λ > 0), which controls the strength of the shrinkage of the variables [30]. We then applied the Lasso feature selection process by shrinking the Lasso regression coefficients of non-informative variables to zero, and selecting the variables of non-zero coefficients. Seven gut microbial species, including *Clostridium bolteae*, *Clostridium ramosum*, *Dorea longicatena*, *Coprococcus eutactus*, *Streptococcus mitis, Clostridiales genomosp. BVAB3*, and *Mobiluncus curtisii* were selected at this step [30]. To reduce overfitting with a limited sample size (*n* = 41), we applied leave-one-out cross-validation (LOOCV) to estimate the prediction performance of BMI loss ratio using a generalized linear model (GLM) of the seven selected features (creatFolds function in R *caret* package and the *glm* function in R *base* package). Likewise, we also used baseline BMI values for LOOCV to estimate its prediction performance for CR-associated BMI loss ratio. Spearman’s rho values were calculated between actual BMI loss ratios and the predicted values.

*P*-value adjustment was applied for multiple hypothesis testing on the concentrations of blood amino acids, the relative abundances of gut microbial genera and species used the Benjamini–Hochberg (BH) method. A BH-adjusted *P* value less than 0.05 was considered as statistically significant. The significance for α-diversity, β-diversity, and phenotypes (age, female to male ratio, BMI, and BMI loss ratio) was set at *p* < 0.05. All statistical analyses were conducted using R (version 3.5.0)

## 3. Results

### 3.1. BMI Loss of ETB and ETP Subjects Responded Differentially to CR Intervention

Based on the baseline genera abundance profile, individuals can be robustly clustered into two enterotypes—enterotype *Bacteroides* (ETB, *n* = 28) and enterotype *Prevotella* (ETP, *n* = 13) (See Materials and Methods, Figure 2A).

Comparisons of the baseline phenotypes between two enterotype groups revealed that all collected phenotypes, including sex distribution (women/men ratio), age and BMI, showed no significant differences between ETB and ETP subjects (Table 2, Figure 2B). By contrast, the baseline compositional and functional characteristics of the gut microbiota showed marked differences between the two groups, in agreement with previous studies [10,31,32]. Thus, genera *Prevotella* and *Paraprevotella* and four species from the two genera were significantly enriched in ETP subjects, whereas 19 common species, from genera *Bacteroides* and *Clostridium*, including *C. bolteae* and *C. ramosum*, were significantly enriched in ETB subjects (Wilcoxon rank-sum test, BH-adjusted *P* < 0.05; Appendix A). At the functional level, multiple pathways were highly enriched in ETB subjects, such as pathways involved in histidine metabolism (map00340), carbohydrate metabolism, secondary bile acid biosynthesis (map00121), membrane transport (phosphotransferase system, map02060; ABC transport, map02010), and in the metabolism of porphyrin and chlorophyll (map00860), lipoic acid (map00785), and biotin (map00780). On the other hand, only five pathways including biosynthesis of phenylalanine, tyrosine and tryptophan (map00400), peptidoglycan (map00550) and terpenoid backbone (map00900), methane metabolism (map00680) and purine metabolism (map00230), were enriched in ETP subjects (|reporter score| > 1.96, Appendix A). These compositional and functional differences between enterotypes might, thus, reflect their microbial trophic niche differentiation.

After the 3-week CR intervention, BMI values were decreased significantly in both ETB and ETP subjects (Figure 2C; Wilcoxon Signed-rank test, *P* < 0.05). Interestingly, subsequent analysis revealed that the ETP subjects showed a significantly greater BMI loss ratio than the ETB subjects (Wilcoxon rank-sum test, *P* < 0.05; mean BMI loss ratio 3.27% versus 1.84%; Figure 2D; see Materials and Methods).

### 3.2. Overall Gut Microbiome Composition Is Stable to CR Intervention

We next investigated the extent and impacts of the CR intervention on gut microbial composition in subjects of different enterotypes. Principal coordinate analysis (PCoA) based on species abundance profile of all samples showed that the projected coordinates of each enterotype group did not change significantly before and after the intervention (Figure 3A, *P* > 0.05). Furthermore, 23 of the 28 ETB subjects and 12 of the 13 ETP subjects were assigned to the same enterotypes after the intervention (Appendix A). In addition, α-diversity (Figure 3B,C) and β-diversity (Figure 3D,E) at the gene and species level of fecal samples also showed no significant changes before and after the intervention in two enterotype groups, respectively (Wilcoxon Signed-rank test, *P* > 0.05). We further revealed that no common species differed significantly in abundance, before and after the intervention in each enterotype group (Appendix A, BH-adjusted *P* > 0.05). All these findings suggest overall stable gut microbial composition in response to a 3-week ~40% energy deficit CR intervention.

### 3.3. Enterotype-Independent Alterations of Blood Amino Acids to CR Intervention

Reflecting differential gut microbial functional potentials including amino acid metabolism between the two enterotype groups (Appendix A), we asked whether blood amino acid composition was associated with enterotype. Principal component analysis (PCA) of baseline amino acid profiles showed no separation of subjects of two enterotypes (Figure 4A). In line with this result, we found no significant differences in the baseline levels of blood amino acids between the two enterotype groups (Wilcoxon rank-sum test, BH-adjusted *P* > 0.05, Figure 4B). We further performed PERMANOVA and identified no significant associations between the changes of blood amino acids and the overall gut microbial composition at baseline (BH-adjusted *P* > 0.05, Appendix A).

We next examined the potential impacts of the CR diet on blood amino acids. Notably, we observed similar changes in multiple blood amino acid concentrations in subjects of both enterotypes, in response to the CR intervention (Figure 4C,D). We, therefore, combined all samples and found that levels of 13 blood AAs and their derivatives such as α-aminoisobutyric acid, β-alanine, serine, glycine, lysine, 2-aminoadipic acid (an intermediate in lysine metabolism), and 3-methyl-histidine were significantly increased whereas only one measured amino acid tyrosine was significantly decreased after the intervention (Wilcoxon Signed-rank test, BH-adjusted *P* < 0.05, Figure 4D). Furthermore, no significant differences were detected in the levels of blood amino acids between ETB and ETP subjects, after the CR intervention (Wilcoxon rank-sum test, BH-adjusted *P* > 0.05, Appendix A), suggesting enterotype-independent effects of the CR intervention on fasting blood amino acids.

### 3.4. Prediction of BMI Loss Ratio Induced by CR Intervention Using Gut Microbial Species

Considering the differential response in the BMI loss ratio to the CR intervention in two enterotypes, we next asked whether we could predict BMI loss ratio from the baseline omics measures. We, thus, built a Lasso (Least absolute shrinkage and selection operator) shrinkage model between baseline levels of gut microbial species and blood amino acids and BMI loss ratio (See Materials and Methods). We successfully selected seven gut microbial species showing associations with BMI loss ratio (absolute coefficient estimate > 0, Figure 5A). Interestingly, the relative abundances of 2 selected species *C. bolteae* and *C. ramosum*, which were enriched in ETB (Wilcoxon rank-sum test, BH-adjusted *P* < 0.05, Appendix A), were negatively correlated with BMI loss ratio. On the other hand, the relative abundance of *D. longicatena*, which was slightly enriched in ETP (Wilcoxon rank-sum test, BH-adjusted *P* = 0.06), was positively correlated with the BMI loss ratio (Figure 5A). Baseline abundances of the other 4 Lasso selected species, however, showed no significant differences between the two enterotype groups (BH-adjusted *P* > 0.05, Appendix A). Among them, the abundances of *Coprococcus eutactus*, *Streptococcus mitis*, and *Clostridiales genomosp. BVAB3* were positively associated with BMI loss ratio, whereas the abundance of *Mobiluncus curtisii* was negatively associated with a BMI loss ratio (Figure 5A).

To estimate the performance of 7 gut microbial species on the prediction of BMI loss ratio, we applied a general linear model between the predicted BMI loss ratios and the actual values using leave-one-out cross-validation (LOOCV). Notably, the result showed a Spearman’s rho of 0.646 between actual and predicted BMI loss ratio, by using baseline abundances of 7 gut microbial species (Figure 5B, see Materials and Methods). By contrast, we found that the individual baseline BMI could hardly predict their BMI loss ratio after the CR intervention, with a Spearman’s rho of −0.016 between the predicted and actual BMI loss ratio (Appendix A).

## 4. Discussion

Little is known about the effect of short-term CR on the gut microbial community and amino acid metabolism in non-obese adult individuals [33]. Here, we conducted a non-controlled study and found that a 3-week 40% energy deficit CR diet could induce a significant reduction in BMI as well as changes in blood amino acids, but limited the changes in gut microbial composition in non-obese subjects. We also demonstrated that CR-induced blood AA changes might be attributed mainly to host skeletal muscle protein breakdown and were gut microbial-independent. Moreover, our study shows that subjects with different baseline enterotypes had differential BMI loss responding to the 3-week CR diet, and pre-intervention gut microbial composition could well predict CR-induced BMI loss.

The levels of multiple AAs and AA derivatives in non-obese individuals were significantly increased after intervention, such as α-aminoisobutyric acid, serine, glycine, lysine, tryptophan, glutamine, aspartate and 3-methylhistidine, a suitable biomarker for skeletal muscle protein breakdown [34,35]. It has been demonstrated that muscle protein degradation is the only endogenous source of 3-methylhistidine in human plasma [36,37]. Although plasma 3-methylhistidine might be influenced by food intake, a study has suggested that about 80% of plasma 3-methylhistidine was of endogenous origin, after an overnight fast [38]. Compared to a regular diet, the amount of daily protein intake was substantially reduced; therefore, the CR diet would hardly contribute to the significant increases in fasting levels of 3-methylhistidine and other amino acids. Based on the above observations, we suggested that the 3-week low energy diet intervention could induce increased host skeletal muscle protein degradation. Interestingly, two previous studies have consistently indicated that compared to CR without exercise, CR with exercise could help to preserve lean mass and have additional beneficial effects on host metabolism [39,40]. Thus, all of the above findings might suggest the need of a combined exercise and CR intervention for reducing host muscle loss resulting from a protein-deficient, low-calorie diet. In contrast, the tyrosine, a nonessential amino acid, which could be found in a variety of protein-rich foods, was significantly reduced after CR intervention, which was in line with the findings from a previous human study, which revealed the impacts of a 2-year long-term dietary intervention on weight loss and circulating amino acids [41].

On the other hand, we showed that a 3-week 40% energy deficit CR diet had limited effects on the microbial alpha- and beta-diversity of non-obese healthy adults, suggesting the overall stability of the adult gut microbiota, in line with a recent study based on 4-week CR intervention in obese women [17]. One possible explanation could be that we reduced the subject’s caloric intake but did not change their long-term eating habits associated with gut microbiota. In addition, some recent studies have indicated that gut microbial responses to a specific diet could be personalized or sex-dependent [42,43]. Further CR intervention studies with larger sample sizes and known food composition are needed to assess individual gut microbiota–diet interactions. Moreover, we observed that ETP subjects exhibited a higher BMI loss ratio than ETB subjects after CR. Consistently, two recent studies have also reported that overweight subjects with a high *Prevotella*/*Bacteroides* ratio had a greater weight loss than those with a low *Prevotella*/*Bacteroides* ratio, when receiving a 6-week whole grain-rich diet [44], or a 24-week energy-deficit diet (~500 kcal/day) [45]. Enterotypes, indeed, have been linked to long-term diet habits [32]. Both *Prevotella* and *Bacteroides* have played pivotal roles in carbohydrate metabolism, while showing various fiber-utilizing capacity [46,47]. The gut microbiota of ETB and ETP subjects, thus, might possibly ferment indigestible dietary fibers from the same diet into different levels and ratios of SCFA (acetate, propionate, and butyrate), which are considered to be energy sources for colonocytes [48]. One possible explanation could be that ETB subjects of higher abundance of *Bacteroides* might have a higher efficiency in energy extraction from a low-calorie diet, to reduce weight loss. Additionally, we also built a microbiota-based model and showed that baseline abundances of seven species showed a high performance for prediction of CR-induced BMI loss. Although none of them were assigned to genera *Prevotella* and *Bacteroides*, a few studies have reported their alterations in obese mice or humans [49,50,51,52]. In line with the higher BMI loss in subjects with ETP, *D. longicatena* was positively associated with BMI loss and enriched in subjects with ETP, while *C. bolteae* and *C. ramosum* were inversely associated with BMI loss and enriched in subjects with ETB. These findings might suggest that clinical implications of non-enterotype-dominant species are still under investigation. In addition to the differences in functional capacity in SCFA production, we and others have shown that different enterotypes also harbored distinct metabolic capacity for several amino acids and secondary bile acids. Compared to type 2 diabetic patients with *Bacteroides* enterotype, those with *Prevotella* enterotype had higher levels of deoxycholic acid and lithocholic acid, and less metabolic improvement, after acarbose treatment [20]. In addition, *Prevotella* could perform biosynthesis of branched-chain amino acids (BCAA), and germ-free mice that received oral gavage of *P. copri* showed elevated plasma BCAA levels and insulin resistance levels [53]. All findings have suggested potential enterotype-dependent differences in modulating host energy and metabolic homeostasis and the underlying mechanisms.

Limitations of the study: We observed that the increase in 3-methylhistidine during CR and suggested that CR induced muscle degradation; however, the measures of fat and lean mass before and after CR were lacking to confirm the inference. We showed that the BMI loss in ETP subjects was higher than that in ETB subjects, but we did not record the regular daily energy intake for each subject. A same calorie-restricted diet might stand for different magnitudes of energy deficits in different subjects. Thus, we could not exclude whether the possible differences in baseline energy consumptions between ETB and ETP subjects could contribute to their differences in BMI loss. We were not able to detect significant associations between gut microbiota and blood AA levels, which might be due to a relatively small sample size of this study. In addition, the study was non-controlled and all volunteers received the same intervention but with no control group, thus, we could not assess the possible short-term variations of gut microbial composition and fasting blood AA profiles in subjects with normal daily calorie intake. We hope that our further work will provide more mechanistic insights into the interactions between CR, and gut microbiota and human metabolism, and will help accelerate the development of microbiota-based personalized solutions in nutrition intervention.

## Figures and Tables

**Figure 1 nutrients-12-00631-f001:**
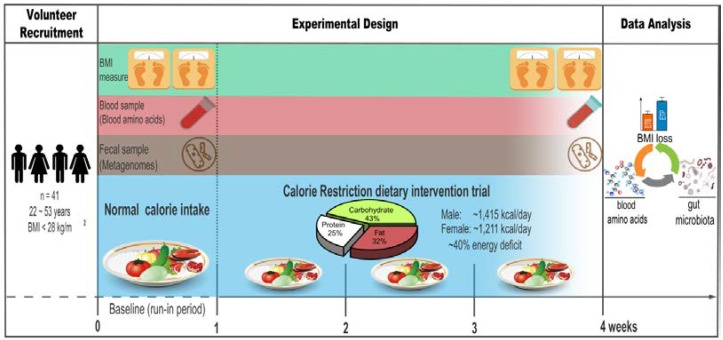
Overview of the experimental design. Illustration of our experimental design, including a 1-week run-in period (baseline) and a 3-week calorie restriction (CR) dietary intervention trial with approximately 40% energy deficit of the recommended daily calorie intake (men, ~1414.9 Kcal/day; women, ~1210.6 Kcal/day). Body Mass Index (BMI), fasting blood samples, and fecal samples of 41 enrolled healthy and non-obese subjects were collected before and after the intervention to assess the effects of CR on BMI, blood amino acids, and the gut microbiome.

**Figure 2 nutrients-12-00631-f002:**
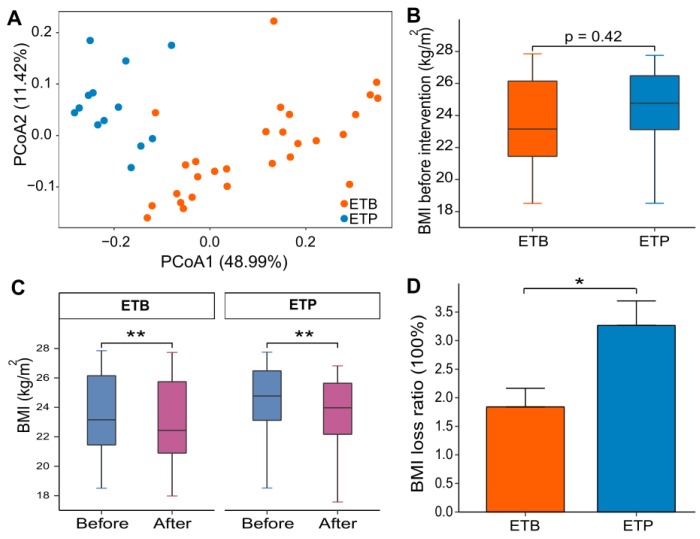
A short-term CR intervention altered BMI. (**A**) Principal coordinates analysis (PCoA) based on genera-level Bray–Curtis distance between all baseline fecal samples. Orange, subjects of enterotype *Bacteroides* (ETB) and blue, subjects of enterotype *Prevotella* (ETP). (**B**) Baseline BMI between ETB and ETP subjects. (**C**) Changes in BMI before and after intervention in individuals of each enterotype. (**D**) Boxplot showing BMI loss ratio between ETB subjects and ETP subjects. *, *P* < 0.05; **, *P* < 0.01.

**Figure 3 nutrients-12-00631-f003:**
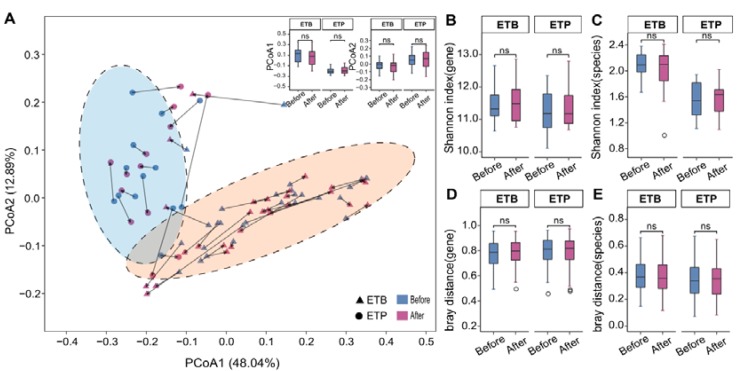
Overall gut microbial composition of two enterotypes before and after the CR. (**A**) Species-based principal coordinates analysis (PCoA) of subjects before and after the CR trial. Triangle, samples of ETB; circles, samples of ETP. Arrows indicate paired samples from the same individual. Boxplot showing the projected coordinate 1 (PCo1) and PCo2 of samples before and after the intervention. (**B**,**C**) α-diversity (Shannon index) at the gene and species levels before (blue) and after (red) intervention in each enterotype group. (**D**,**E**) β-diversity (Bray–Curtis distance) at the gene and species levels, before (blue) and after (red) the intervention in each enterotype group. ns—no significance, *P* > 0.05, Wilcoxon signed-rank test.

**Figure 4 nutrients-12-00631-f004:**
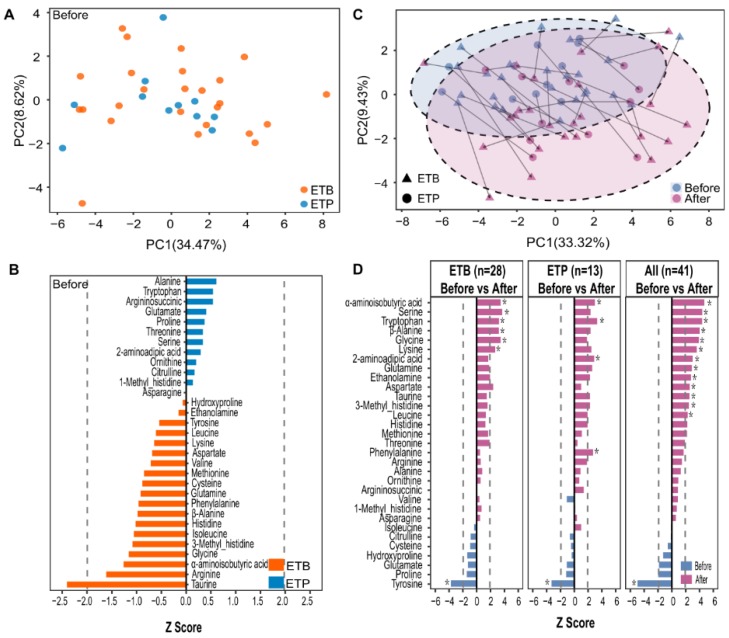
A short-term CR intervention altered the blood amino acids. (**A**) Principal component analysis (PCA) of 41 subjects using baseline blood amino acid profiles. Orange, ETB; blue, ETP. (**B**) Comparison of baseline blood amino acid levels between ETB and ETP subjects; dashed lines indicate the absolute Z score of 1.96 (*P* = 0.05); orange and blue bars indicate that the Z-score of blood amino acid was overrepresented in ETB and ETP subjects, respectively; Wilcoxon rank-sum test, *P* values are transformed to Z-scores to represent enrichment directions. (**C**) Amino acid-based PCA of samples before and after the intervention. Triangle—samples of ETB subjects; Circles—samples of ETP subjects. Arrows indicate paired samples from the same individual. (**D**) Changes in blood amino acid concentrations of ETB subjects, ETP subjects, and all subjects before and after the intervention. Wilcoxon signed-rank test, *P* values are transformed to Z-scores to represent enrichment directions. Dashed lines indicate the absolute Z score of 1.96 (*P* = 0.05). Asterisk (*) indicates the statistical significance at Benjamini–Hochberg (BH) adjusted *P* < 0.05.

**Figure 5 nutrients-12-00631-f005:**
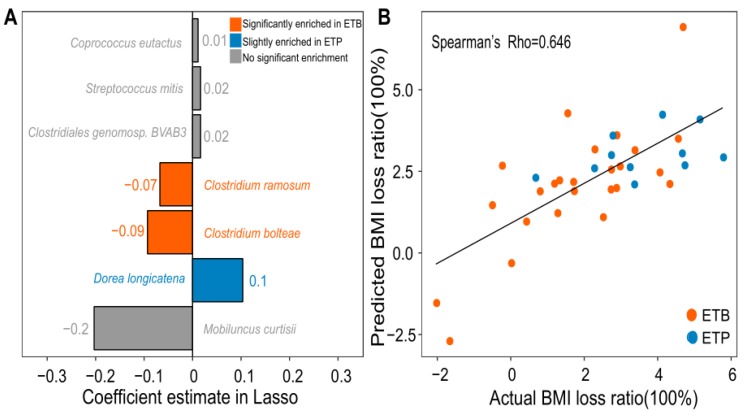
Prediction of BMI loss ratio using baseline abundances of gut microbial species. (**A**) Bar plot showing the 7 gut microbial species selected by least absolute shrinkage and selection operator (Lasso). Bar length indicates a regression coefficient of each species estimated by Lasso. Orange, species significantly enriched in ETB subjects (BH-adjusted *P* < 0.05); blue, species slightly enriched in ETP subjects (*P* < 0.05 and BH-adjusted *P* = 0.06); grey, species with no significant enrichment between two enterotypes (*P* > 0.05). (**B**) Scatter plot showing prediction performance of BMI loss ratio based on the 7 selected species. Leave-one-out cross-validation (LOOCV) was applied to evaluate the performance of the generalized linear model (GLM), showing a strong Spearman’s rho between actual BMI loss ratios and predicted BMI loss ratios of 0.646. Red circles—ETB individuals; blue circles—ETP individuals.

**Table 1 nutrients-12-00631-t001:** Cohort Description.

Cohort	Total (Mean ± SD)	Women	Men	*P* Value (Women vs. Men)
**Number of subjects**	41	24	17	/
**Age**	30 ± 6	28 ± 5	32 ± 9	0.383
**BMI (kg/m^2^)**	23.72 ± 2.81	23.31 ± 2.50	24.30 ± 3.18	0.198

**Table 2 nutrients-12-00631-t002:** Comparison of baseline phenotypes of ETB and ETP subjects.

	ETB Group (Mean ± SD)	ETP Group (Mean ± SD)	*P* Value ETB vs. ETP
**Number of subjects**	28	13	
**Sex (women/men)**	16/12	8/5	1
**Age**	29 ± 6	30 ± 7	0.44
**BMI (kg/m^2^)**	23.50 ± 2.81	24.21 ± 2.85	0.42

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
