# Peer review of "Effect of Caloric Restriction on BMI, Gut Microbiota, and Blood Amino Acid Levels in Non-Obese Adults"

_nutrients, 2020, doi:10.3390/nu12030631_

Round 1
Reviewer 1 Report
Please find enclosed my comments concerning the manuscript entitled: “Effect of caloric restriction on BMI, gut microbiota, and blood amino acid levels in non obese adults. 

I have still some comments aimed to improve the information given in the paper before being published.
Abstract. It must be improved. It is difficult to understand the purpose of this study: why this study has been designed only on non obese adults? Introduction. It's hard to find your way around all the microbial metabolites you mentioned. May be better able to synthesize the information Materials and methods. Please detail more precisely the cohort description. I don’t understand why you have polled males and females Results/Discussion. Many information are presented in supplemental data; however, it is difficult to find relevant information in these tables. It will be interested to integrate the results in function of KEGG pathways. I think presenting and interpreting results between non obese males and females will be more relevant.
Concerning figures, they are well presented.
Reviewer 2 Report
Reviewer’s comments and suggestions for Authors
In this manuscript, the author wants to assess the effect of a short-term caloric restriction (CR) diet on various parameters such as BMI, blood amino acids (AAs) and the gut microbiota in healthy nonobese subjects. The study was investigated in 3-week CR intervention in 41 subjects, with roughly 40% less of calories than the usual daily energy intake. They measured BMI, and applied targeted metabolic profiling on fasting blood samples and shotgun metagenomic sequencing on fecal samples before and after intervention.
The result of the study reported that CR decreased BMI in most subjects, and Prevotella (ETP) subjects exhibited a significantly higher BMI loss ratio than the Bacteroides (ETB) subjects. Moreover, CR induced limited changes in gut microbial composition, but substantial microbial-independent changes in blood AAs, including a significant increase in 3 methylhistidine.
The manuscript is in good shape but there were still typo errors and ambiguous sentences to be incorporated in the manuscript.
Nonetheless, the manuscript required to improve by reviewing a few minor concerns.
It would be better if the author chooses an equal number of participants in two enterotypes on the basis of statistical significance. Line 44-45, the line is confusing to understand. Make it simple statement the deterioration of obesity effect on the body. Line 53-54, is they provide any reason for that, you need to mention here. Line 59-60, it is an ambiguous line, needs to change it. Line 65, small a is needed in place of A of acarbose. Line 78, On the contrary, baseline amino acid profiling showed written by the author means “baseline blood amino acid profiling”. It needed to clear Line 84, no need to write reference here Line 100, no information about the weekend food Line 119-120, the author should have to write the percentage of all nutrients not only carbohydrate and protein Line 275-276, what would be the reason for this result Line 310, What is the meaning of individual baseline BMI here, in above sentence they said correlation The discussion should be rewritten more as the results were not fully described and discussed here in this section. Line 372-373, need explain more comprehensively Most of the references are not according to the journal guidelines, please check reference number 2,5,7,8,10, 26,36, 41.
Round 2
Reviewer 1 Report
the answers provided by the authors are satisfactory and the changes made have greatly improved the manuscript and I agree for publication of this article.
Author Response
Thanks very much for your kind comments